# Studying the biology of cytotoxic T lymphocytes in vivo with a fluorescent granzyme B-mTFP knock-in mouse

Praneeth Chitirala[1†], Hsin-Fang Chang[1†], Paloma Martzloff[1], Christiane Harenberg[2], Keerthana Ravichandran[1], Midhat H Abdulreda[3,4,5,6], Per-Olof Berggren[3,4,7,8], Elmar Krause[1], Claudia Schirra[1], Trese Leinders-Zufall[9], Fritz Benseler[2], Nils Brose[2], Jens Rettig[1*]

[1]Cellular Neurophysiology, Center for Integrative Physiology and Molecular Medicine (CIPMM), Saarland University, Homburg, Germany; [2]Department of Molecular Neurobiology, Max-Planck-Institute of Experimental Medicine, Göttingen, Germany; [3]Diabetes Research Institute and Cell Transplant Center, University of Miami Miller School of Medicine, Miami, United States; [4]Department of Surgery, University of Miami Miller School of Medicine, Miami, United States; [5]Department of Microbiology and Immunology, University of Miami Miller School of Medicine, Miami, United States; [6]Department of Ophthalmology, University of Miami Miller School of Medicine, Miami, United States; [7]Diabetes Research Institute Federation, Hollywood, United States; [8]The Rolf Luft Research Center for Diabetes and Endocrinology, Karolinska Institutet, Karolinska University Hospital, Stockholm, Sweden; [9]Sensory and Neuroendocrine Physiology, Center for Integrative Physiology and Molecular Medicine (CIPMM), Saarland University, Homburg, Germany

*For correspondence:
jrettig@uks.eu

[†]These authors contributed equally to this work

**Abstract** Understanding T cell function in vivo is of key importance for basic and translational immunology alike. To study T cells in vivo, we developed a new knock-in mouse line, which expresses a fusion protein of granzyme B, a key component of cytotoxic granules involved in T cell-mediated target cell-killing, and monomeric teal fluorescent protein from the endogenous *Gzmb* locus. Homozygous knock-ins, which are viable and fertile, have cytotoxic T lymphocytes with endogenously fluorescent cytotoxic granules but wild-type-like killing capacity. Expression of the fluorescent fusion protein allows quantitative analyses of cytotoxic granule maturation, transport and fusion in vitro with super-resolution imaging techniques, and two-photon microscopy in living knock-ins enables the visualization of tissue rejection through individual target cell-killing events in vivo. Thus, the new mouse line is an ideal tool to study cytotoxic T lymphocyte biology and to optimize personalized immunotherapy in cancer treatment.

## Introduction

Cytotoxic T lymphocytes (CTLs) are not only essential for the removal of foreign agents such as viruses or bacteria, but also play a key role in modern personalized cancer immunotherapy (*Porter et al., 2011*; *Sharma and Allison, 2015*; *Watanabe et al., 2018*; *Minutolo et al., 2019*). Accordingly, a detailed mechanistic understanding of the major CTL function, that is the killing of cells through release of cytotoxic substances from cytotoxic granules (CGs) at the immune synapse (IS), is of utmost interest for basic and clinical science alike (*Dustin and Long, 2010*; *Griffiths et al., 2010*; *Mukherjee et al., 2017*; *Xiong et al., 2018*). The release of perforin and granzymes from

**eLife digest** Cytotoxic, or killer, T cells are a key part of the immune system. They carry a lethal mixture of toxic chemicals, stored in packages called cytotoxic granules. Killer T cells inject the contents of these granules into infected, cancerous or otherwise foreign cells, forcing them to safely self-destruct. In test tubes, T cells are highly efficient serial killers, moving from one infected cell to the next at high speed. But, inside the body, their killing rate slows down. Researchers think that this has something to do with how killer T cells interact with other immune cells, but the details remain unclear.

To get to grips with how killer T cells work in their natural environment, researchers need a way to follow them inside the body. One approach could be to use genetic engineering to attach a fluorescent tag to a protein found inside killer T cells. That tag then acts as a beacon, lighting the cells up and allowing researchers to track their movements. Tagging a protein inside the cytotoxic granules would allow close monitoring of T cells as they encounter, recognize and kill their targets. But fluorescent tags are bulky, and they can stop certain proteins from working as they should.

To find out whether it is possible to track killer T cells with fluorescent tags, Chitirala, Chang et al. developed a new type of genetically modified mouse. The modification added a teal-colored tag to a protein inside the granules of the killer T cells. Chitirala, Chang et al. then used a combination of microscopy techniques inside and outside of the body to find out if the T cells still worked. This analysis showed that, not only were the tagged T cells able to kill diseased cells as normal, the tags made it possible to watch it happening in real time. Super-resolution microscopy outside of the body allowed Chitirala, Chang et al. to watch the killer T cells release their toxic granule content. It was also possible to follow individual T cells as they moved into, and destroyed, foreign tissue that had been transplanted inside the mice.

These new mice provide a tool to understand how killer T cells really work. They could allow study not only of the cells themselves, but also their interactions with other immune cells inside the body. This could help to answer open questions in T cell research, such as why T cells seem to be so much more efficient at killing in test tubes than they are inside the body. Understanding this better could support the development of new treatments for viruses and cancer.

CGs has been studied in great detail using cultured primary CTLs (*Friedl et al., 2005*; *Kupfer, 2006*; *Orange, 2008*; *Dustin and Long, 2010*; *Griffiths et al., 2010*; *Voskoboinik et al., 2015*), but corresponding approaches have neglected the fact that CTLs act in concert with other immune cells to perform their function in vivo. In essence, CTL function in isolation differs considerably from CTL function in vivo. However, in vivo studies on CTLs have been scarce due to their technical difficulties and the lack of suitable markers (*Mempel et al., 2006*; *Breart et al., 2008*; *Germain et al., 2012*; *Halle et al., 2017*; *Lodygin and Flügel, 2017*; *Torcellan et al., 2017*; *Cazaux et al., 2019*; *Malo and Hickman, 2019*).

We report here the generation of a GzmB-mTFP knock-in (KI) mouse that expresses a fluorescent fusion protein consisting of granzyme B (GzmB), a CG-resident serine protease, and monomeric teal fluorescent protein (mTFP) from the endogenous *Gzmb* locus. The new GzmB-mTFP-KI allows the observation of individual CTLs and even CGs in living mice at any time point of interest. We show that GzmB-mTFP-KIs are viable, fertile and free of any obvious defects, that their T cell-specific functions are wild-type-identical, and that their CTLs can be imaged with all major super-resolution techniques in vitro and in vivo. We expect that the GzmB-mTFP-KI will be a highly valuable tool to investigate CTL function in vitro and in vivo - in the context of both, basic CTL biology and clinical aspects of CTL function, such as CTL-based personalized cancer immunotherapy.

## Results

### Generation of a GzmB-mTFP-KI mouse line

To create a specific, endogenous fluorescent label for cytotoxic granules (CG) we chose GzmB (*Young et al., 1986*; *Masson and Tschopp, 1987*; *Krahenbuhl et al., 1988*), which belongs to a family of serine proteases that induce apoptosis of target cells and which is present in CGs of natural

killer cells and CD4[+] and CD8[+] T lymphocytes (*Peters et al., 1991*). In contrast to perforin, a CG-specific pore-forming protein, GzmB deletion does not lead to a killing defect in CTLs (*Simon et al., 1997*). Using CRISPR-Cas9 technology and a corresponding HDR fragment, we replaced the Stop codon in exon 5 of the mouse *Gzmb* gene with a sequence encoding a flexible GGSGGSGGS linker, which has a high probability to be cleaved in the acidic environment of the lysosome (*Huang et al., 2014*), the coding sequence of monomeric teal fluorescent protein (mTFP), and a Stop codon (*Figure 1A* and *Figure 1—figure supplement 1*). We generated homozygous GzmB-mTFP-KIs, which were viable and fertile and showed no obvious phenotypic changes. PCR analyses of CTL lysates derived from wild-type, heterozygous and homozygous GzmB-mTFP-KI mice verified the expected genotypes (*Figure 1B*). As envisioned by our design, Western blot analyses of lysates of CTLs four and five days after activation showed that the fusion protein is efficiently cleaved into GzmB and mTFP (*Figure 1C*), ensuring a correct function of GzmB in the killing process. As expected, Western blot (days 0–5; *Figure 1D*) and FACS analyses (days 0–10; *Figure 1E*) demonstrated a continuous up-regulation of GzmB expression following CTL activation. The expression levels of the fusion protein varied between different preparations (59.1% (day 4, *Figure 1C*), 53.6% (day 5, *Figure 1C*) and 183.9% (day 5, *Figure 1D*) of wt level for GzmB) as expected, but we always observed a robust fluorescence without the requirement to change the intensity of the excitation lasers for the experiments shown in the following figures.

## Endogenous GzmB-mTFP fluorescence co-localizes with CG markers and migrates to the IS

To verify the usefulness of the GzmB-mTFP-KI in vitro and in vivo imaging, we next assessed the correct localization of GzmB and mTFP to CGs. We isolated CTLs from the KI mice and fixed them five days after activation. We then performed immunofuorescence labeling with anti-GzmB and anti-perforin antibodies and detected by structured illumination microscopy (SIM) an excellent co-localization of mTFP, GzmB, and perforin signals (Pearson's coefficient of correlation 0.84 ± 0.02 for GzmB-mTFP vs. GzmB, 0.69 ± 0.03 for GzmB-mTFP vs. perforin, n = 10; Manders' coefficient of correlation 0.79 ± 0.02 for GzmB-mTFP vs. GzmB, 0.80 ± 0.01 for GzmB vs. GzmB-mTFP, 0.70 ± 0.02 for GzmB-mTFP vs. perforin, 0.71 ± 0.03 for perforin vs. GzmB-mTFP, n = 10; *Figure 2A–C*).

Further, we cross-bred the GzmB-mTFP-KI with a Synaptobrevin-2 (Syb2)-mRFP-KI. Using the Syb2-mRFP-KI, we had shown previously that Syb2 is almost exclusively localized to CGs in CTLs and acts as the vSNARE in CG fusion with the plasma membrane (*Matti et al., 2013*). Comparison of endogenous GzmB-mTFP and Syb2-mRFP fluorescence in CTLs on poly-ornithine or anti-CD3 coated coverslips again revealed excellent co-localization (Pearson's coefficient of correlation 0.88 ± 0.10 for poly-ornithine, and 0.86 ± 0.08 for anti-CD3, n = 25; Manders' coefficient of correlation 0.86 ± 0.08 for GzmB-mTFP vs. Syb2-mRFP and 0.79 ± 0.17 for Syb2-mRFP vs. GzmB-mTFP on poly-ornithine, 0.79 ± 0.13 for GzmB-mTFP vs. Syb2-mRFP and 0.81 ± 0.13 for Syb2-mRFP vs. GzmB-mTFP on anti-CD3, n = 25; *Figure 2D–F*). These data show that the cleavage of GzmB-mTFP we observed (*Figure 1C–D*) occurs inside CGs, so that the mTFP in the GzmB-mTFP-KI represents an excellent endogenous marker for studies of CG maturation, transport and fusion, without affecting GzmB localization and function. To visualize CG transport along the microtubular network, we incubated primary CTLs from GzmB-mTFP-KIs with silicon rhodamine(SiR)-tubulin, seeded them onto anti-CD3 coated coverslips to induce immune synapse (IS) formation and followed transport of CGs along microtubuli by live stimulated emission depletion (STED) microscopy (*Figure 2G*). Within five minutes after seeding, microtubules with the characteristic microtubule organizing center (MTOC) had formed and all CGs were being transported toward the IS at the bottom of the cells. During transport, the average distance between CGs and microtubuli was 137.5 ± 19.6 nm (19 cells, 80 granules; *Figure 2G–I*), comparable to that of wild-type control cells. Thus, CTLs of the GzmB-mTFP-KI mouse are ideally suited to follow maturation and trafficking of endogenously labeled CGs in vitro with super-resolution microscopy.

## GzmB-mTFP containing CGs fuse at the immune synapse

We next tested whether CG fusion at the IS can be observed and quantified in real time using the GzmB-mTFP-KI. For this purpose, we seeded primary GzmB-mTFP-KI CTLs on anti-CD3 coated coverslips and performed total internal reflection microscopy (TIRFM). The presence of essential

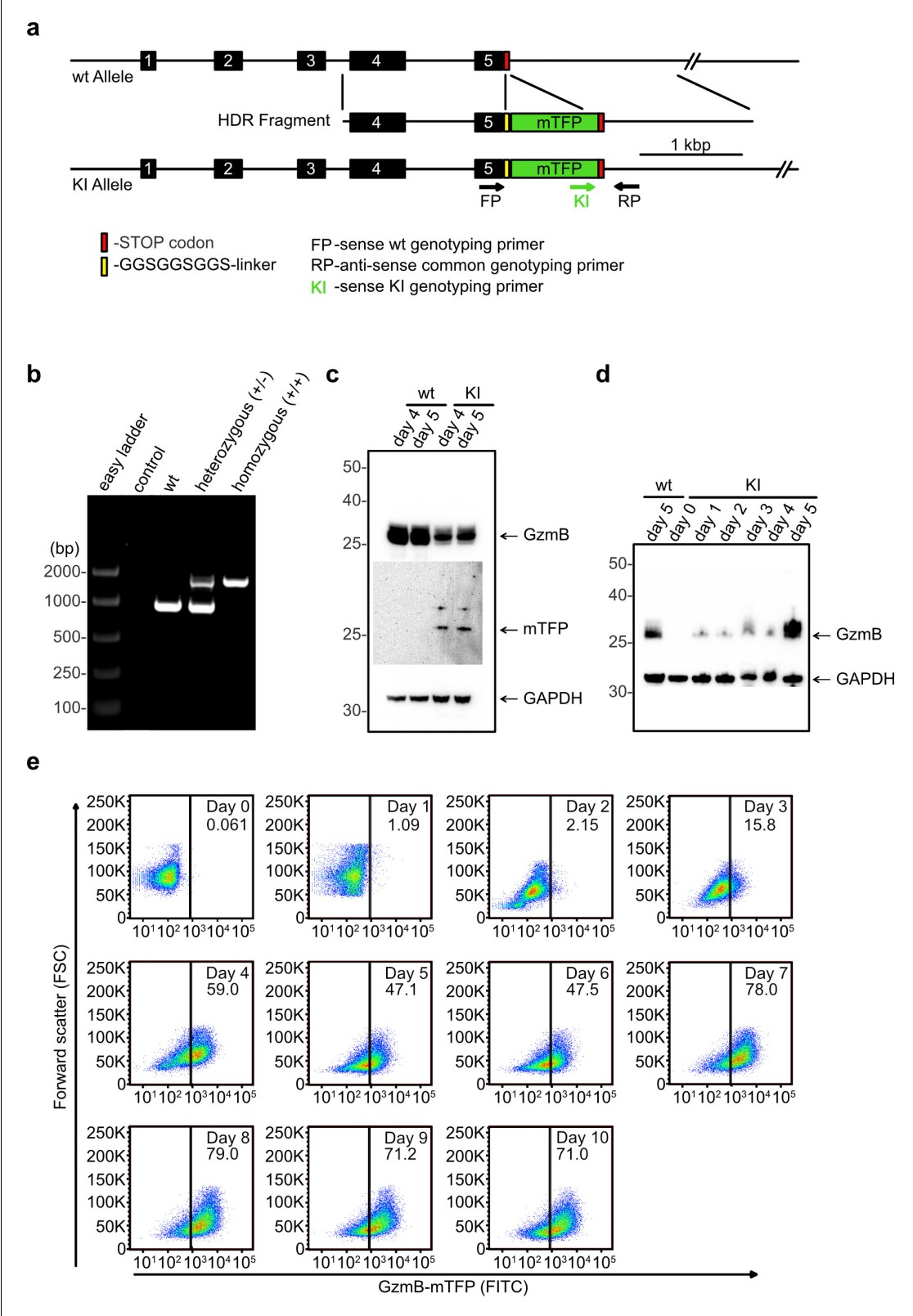

**Figure 1.** Generation of GzmB-mTFP knock-in mice. (**A**) CRISPR-Cas9 strategy to generate the GzmB-mTFP-KI. wt, wild-type; KI, GzmB-mTFP-KI; numbered black boxes, *Gzmb* exons; red bar, Stop codon; yellow bar, GGSGGSGGS-linker; green box, mTFP coding sequence; rightward black arrow, forward genotyping primer wt; rightward green arrow, forward genotyping primer KI; leftward black arrow, reverse common genotyping primer (primers are not drawn to scale). (**B**) PCR of CTL lysates derived from wild-type, heterozygous and homozygous GzmB-mTFP-KI mice using oligonucleotides FP,

*Figure 1 continued on next page*

*Figure 1 continued*

RP and KI. (C) Western blot of lysates derived from wild-type and GzmB-mTFP-KI CTLs 4 and 5 days after activation. Anti-GzmB and anti-mTFP antibodies were used for detection, anti-GAPDH antibody served as loading control. (D) Western blot of lysates derived from naïve GzmB-mTFP-KI CTLs and 1, 2, 3, 4 and 5 days after activation with anti-CD3/anti-CD28 coated beads. Lysates from wild-type CTLs 5 days after activation were used for comparison, anti-GAPDH antibody served as loading control. (E) CTLs from GzmB-mTFP-KI mice were isolated and analyzed by FACS at the indicated days after activation. Non-activated CTLs (day 0) served as negative control.

The online version of this article includes the following figure supplement(s) for figure 1:

**Figure supplement 1.** Design of the HDR fragment to generate the GzmB-mTFP-KI The HDR fragment was designed to replace the endogenous Stop codon of the *Gzmb* gene by sequences encoding a GGSGGSGGS-linker, mTFP and a Stop codon.

components of pSMAC and cSMAC allows the CTLs to form an IS on the area facing the coverslip. Within two minutes CGs arrived at the IS and frequently fused, as indicated by a sudden drop in fluorescence due to the diffusion of the fluorophore (*Figure 3A*). Quantification showed that the average number of CTLs showing CG fusion (59.0 ± 2.9% for wild-type control and 57.7 ± 4.6% for KI, N = 3, n = 70; *Figure 3B*) and the number of CG fusion events per cell (7.09 ± 0.48 for wild-type control and 7.23 ± 0.64 for KI, N = 3, n = 70; *Figure 3C*) were indistinguishable between wild-type and GzmB-mTFP-KI CTLs, in accordance with our recent findings on fusion kinetics and TCR stimulation (*Estl et al., 2020*). Furthermore, there was no difference in the kinetics of CG movement into the TIRF field between wild-type and GzmB-mTFP-KI (*Figure 3D*).

We also used CTLs isolated from GzmB-mTFP/Syb2-mRFP double KI mice to assess simultaneous fluorescence signals in mTFP and mRFP channels. Again, we found a perfect co-localization of both fluorophores, with fusion events detected in both channels. Following fusion, the mRFP fluorescence remained visible at the plasma membrane, as expected for a membrane protein like synaptobrevin2 (*Figure 3E*). Our TIRFM data show that CG fusion events at the IS can be reliably observed with CTLs derived from the GzmB-mTFP-KI mouse, making the new KI an ideal tool to study molecular mechanisms of CG fusion in primary CTLs without the need for transfection of CG markers.

## GzmB-mTFP-KI CTLs efficiently kill target cells

To confirm that primary GzmB-mTFP-KI CTLs maintain their ability to kill target cells, we performed two independent killing assays and compared the killing efficiency of CTLs from GzmB-mTFP-KIs to that of wild-type CTLs. First, we stably transfected P815 target cells with Casper3-GR, a FRET based sensor consisting of green and red fluorescent proteins TagGFP and TagRFP connected by a linker containing the caspase-3 cleavage sequence DEVD (*Shcherbo et al., 2009*). The activation of caspase-3 by GzmB during apoptosis leads to cleavage of the DEVD sequence and elimination of FRET, which can be detected as a decrease in the red emission of TagRFP and a simultaneous increase in green emission of TagGFP. We found that wild-type and GzmB-mTFP-KI CTLs killed with a similar frequency (59.0 ± 2.9% for wild-type and 57.7 ± 4.6% for KI, N = 3, n = 70) and efficiency, as indicated by the same fold ratio change between the green and the red channel (2.40 ± 0.33 for wild-type, n = 8; 2.34 ± 0.22 for KI, n = 7; *Figure 4A*).

Second, we performed a flow cytometry-based degranulation assay (*Betts et al., 2003*). At all three days after activation tested (days 3, 5 and 7 after activation), the degranulation measured by LAMP1 (CD107a) expression on the surface was comparable between wild-type and GzmB-mTFP-KI CTLs [day 3: 84.0 ± 2.5% (wild-type) vs. 82.7 ± 0.3% (KI); day 5: 84.2 ± 2.5% (wild-type) vs. 82.6 ± 0.3% (KI); day 7: 88.3 ± 0.6% (wild-type) vs. 85.4 ± 0.2% (KI), N = 3; *Figure 4B*). Thus, the killing capacity of GzmB-mTFP-KI CTLs is indistinguishable from that of wild-type CTLs in vitro.

## In vivo imaging of allorejection using GzmB-mTFP-KI CTLs

CTLs fight infections and cancer in vivo in concert with other immune cells, like CD4[+] T cells and macrophages. Therefore, we tested the ability of GzmB-mTFP-KI CTLs to reject foreign tissue transplanted into the anterior chamber of the eye (ACE) (*Speier et al., 2008a*; *Speier et al., 2008b*; *Abdulreda et al., 2013*). Originally developed to follow insulin secretion from Langerhans islets in Diabetes treatment (*Speier et al., 2008a*; *Rodriguez-Diaz et al., 2012*), the ACE model allows longitudinal, non-invasive in vivo imaging of anti-islet immune responses with single-cell resolution in real time (*Abdulreda et al., 2013*; *Abdulreda et al., 2019*). We transplanted Langerhans islets from

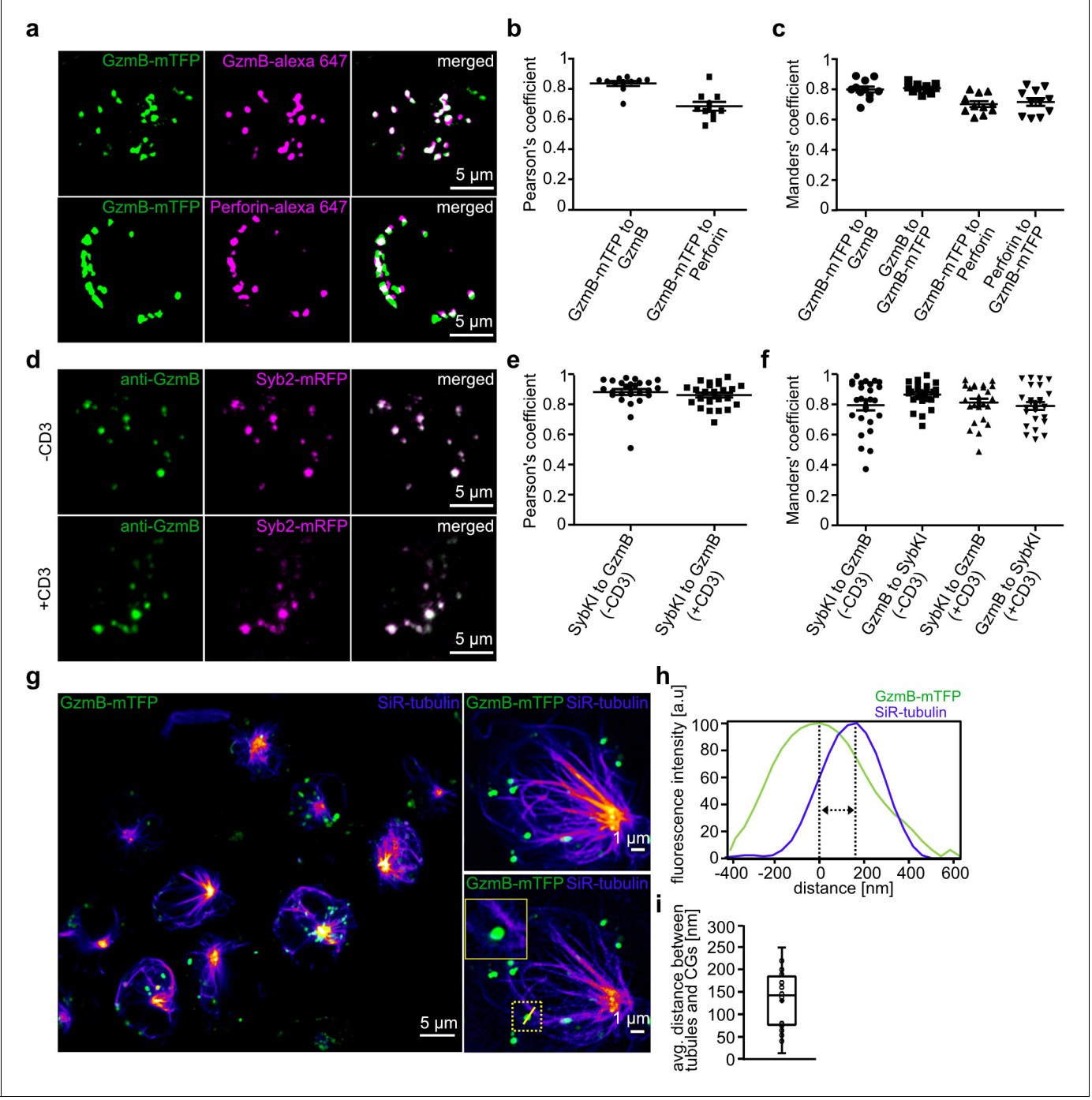

**Figure 2.** Endogenous GzmB-mTFP fluorescence co-localizes with CG markers and migrates to the IS. (A) SIM images of primary CTLs from GzmB-mTFP-KI mice on day five after activation. CTLs were fixed and stained with Alexa647 conjugated anti-GzmB antibody (upper panel) or Alexa647 conjugated anti-perforin antibody (lower panel). (B–C) Pearson's and Manders coefficients of corelation between GzmB-mTFP and the other CG markers. (D) SIM images of primary CTLs from Syb2-mRFP/GzmB-mTFP double KI mice on day five after activation. CTLs were fixed and examined for endogenous fluorescence in the mTFP and mRFP channels. (E–F) Pearson's and Manders coefficients of correlation between GzmB-mTFP and Syb2-mRFP. Scale bars, 5 μm. (G) Live STED images of primary CTLs from GzmB-mTFP-KI mice labeled with SiR-tubulin and plated onto anti-CD3 coated coverslip. The transport of CG along the microtubules toward the IS is shown in the overview for all CTLs (left; Scale bar, 5 μm) and in more detail in the two exemplary CTLs (right; Scale bar, 1 μm). (H) Distance between CG and microtubule from exemplary image quantified by ImageJ software. (I) Average distance between microtubules and CGs (n = 19).

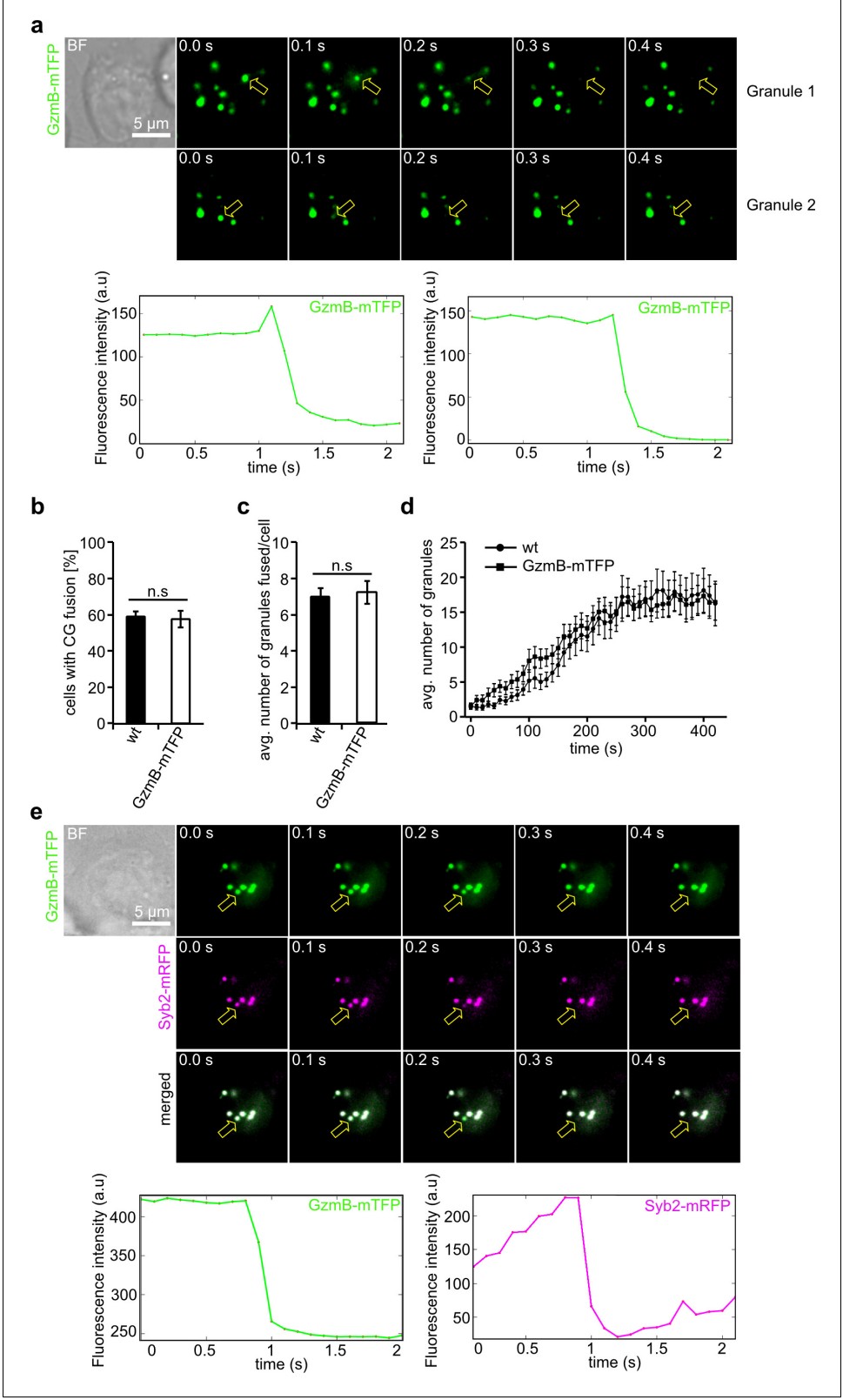

**Figure 3.** GzmB-mTFP containing cytotoxic granules fuse at the IS. (A) Selected live-cell TIRF microscopy images of GzmB-mTFP in a CTL in contact with an anti-CD3 coated coverslip. Fusion events are indicated with open arrows (seven consecutive frames per fusion event are shown). Scale bar, 5 µm. (B) Selected live-cell, dual-channel TIRF microscopy images of GzmB-mTFP and Syb2-mRFP in a CTL derived from Syb2-mRFP/GzmB-mTFP double KI

*Figure 3 continued on next page*

*Figure 3 continued*

mice and in contact with an anti-CD3 coated coverslip. Fusion events are indicated with open arrows (seven consecutive frames per fusion event are shown). (**C–D**) Comparison between wild-type and GzmB-mTFP-KI CTLs reveals no difference in the percentage of CTLs showing fusion (**C**) and average number of fused CG per cell (**D**). (**E**) Mean average number of CGs appearing in the TIRF plane per cell during 7 min of measurements (N = 3, n = 10 for wt; and N = 3, n = 12 for GzmB-mTFP KI).

allogenic DBA/2 donor mice into the anterior chamber of the eye of GzmB-mTFP-KIs recipients, which had been bred in the C57BL/6 background (*Figure 5A*). At post operational day 4 (POD4) following transplantation, we started weekly in vivo observations with a confocal laser scanning or a two-photon microscope. Between POD4 and POD7, transplanted islets became capillarized and blood-perfused, indicating successful engraftment (*Figure 5—figure supplement 1A*, *Video 1*). Starting between POD7 and POD14, GzmB-mTFP positive cells invaded the islets, reached a maximum in number within two weeks (26 ± 8.3 cells/100 µm$^3$, N = 4 animals, n = 11 islets) and eventually declined with the disappearance of the island allografts (*Figure 5—figure supplement 1B–D*). To exclude that the islet shrinking resulted from islet necrosis and/or unspecific rejection, we performed control experiments with transplanted syngenic islets from C57BL/6 donors. As expected, we observed engraftment of the syngenic islets after 4–7 days, but we did not detect any GzmB-mTFP positive cells nor any islet rejection during three months of observation post-transplantation, demonstrating the high specificity of the ACE model.

To verify the identity of GzmB-mTFP positive cells inside the anterior chamber, we fixed transplanted eyes at POD14-18, produced histochemical samples and counterstained the tissue with

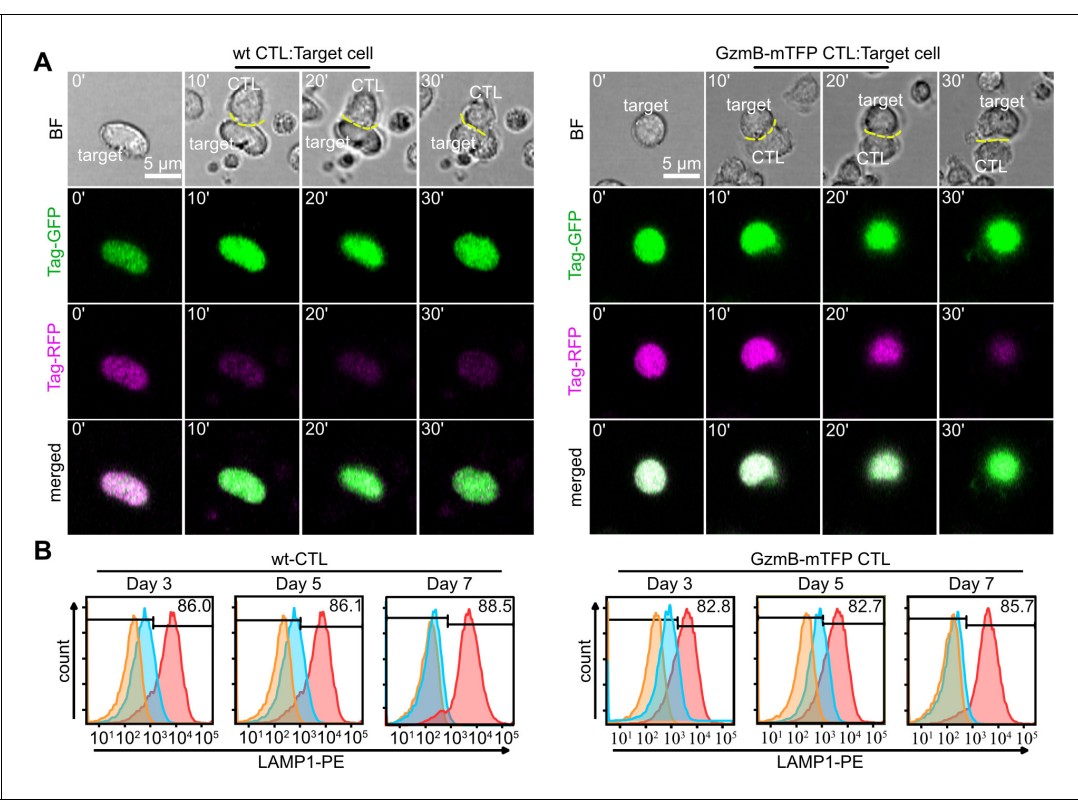

**Figure 4.** CTLs from GzmB-mTFP-KI mice kill target cells as efficient as wild-type CTLs. (**A**) Live cell killing assay showing a wild-type (left) and GzmB-mTFP-KI (right) CTL in contact with P815 target cells stably expressing Casper3-GR (FRET construct containing Tag-GFP and Tag-RFP with a target cleavage site DEVD of Caspase 3 (activated via GzmB). LSM images at indicated time points show a reduction in FRET signal, indicating the killing of the target cell. Scale bars, 5 µm. (**B**) CTLs isolated from either wt (left) or GzmB-mTFP-KI (right) mice were assessed for degranulation capacity using FACS-based assay on day 3, 5 and 7, respectively. Data shown are representative histograms from three independent experiments. Orange areas show untreated CTLs, blue areas show constitutive secretion and red areas show stimulated secretion.

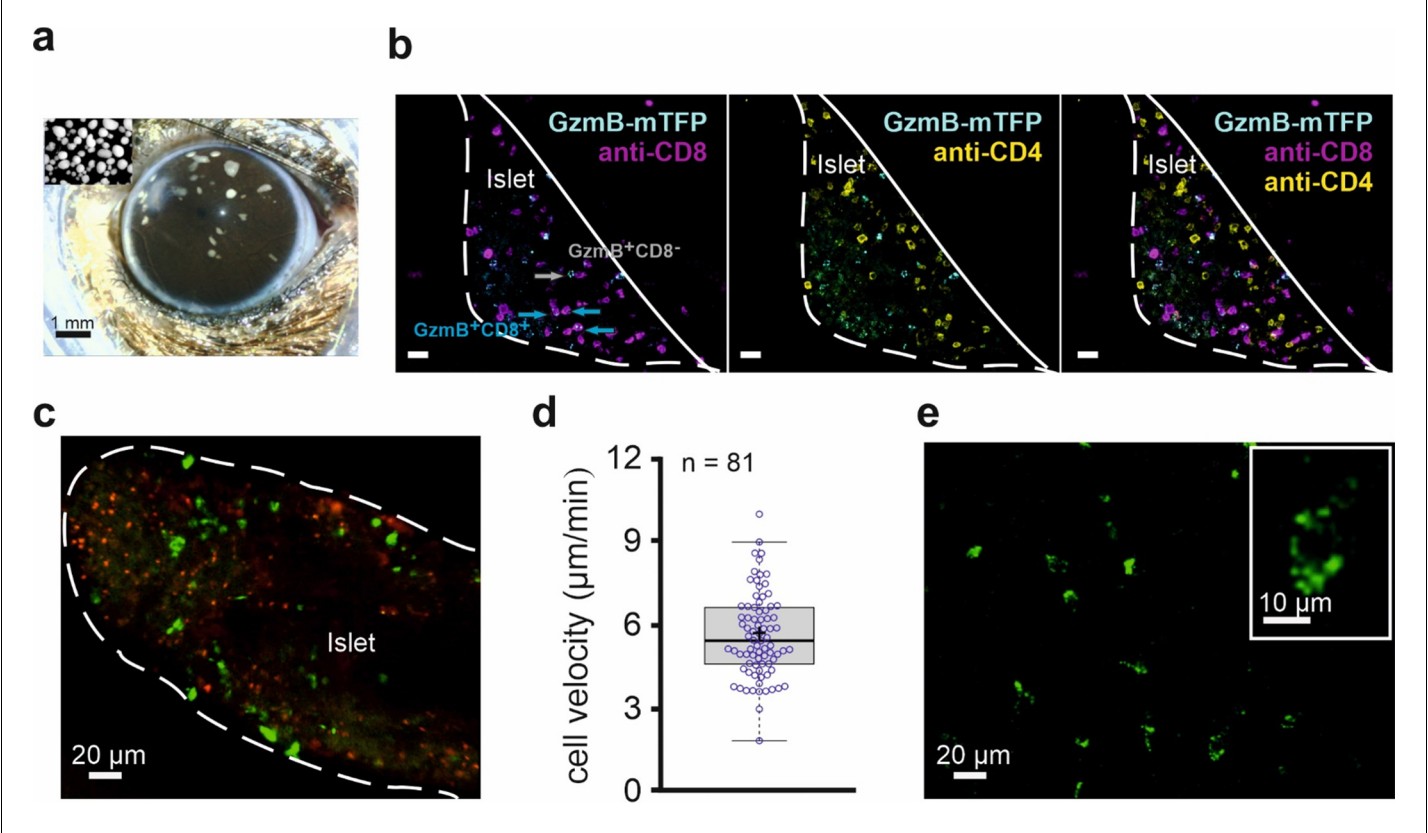

**Figure 5.** In vivo imaging of allorejection in the anterior chamber of the eye (ACE) visualized by CTLs from GzmB-TFP-KI mice. (**A**) Binocular image of a mouse eye with islets of Langerhans from DBA/2 mice transplanted into the anterior chamber of the eye (ACE) 4 days prior to the image. Islets can be identified as small white spots in the range between 100–300 µm diameter located on top of the iris. Inset shows cultured islets before transplantation. (**B**) Confocal images of a pancreatic islet inside the ACE fixed and sliced at POD14. The outline defines the boundary of the islet of Langerhans localize on the iris. Green cells are cytotoxic T lymphocytes marked by endogenously expressed GzmB-mTFP having infiltrated the islet during the immune response. Antibody staining of CD8+ (blue) and CD4+ (magenta) T cells are shown. (**C**) In vivo confocal microscopy of an islet during allorejection in the ACE (POD14). White line marks the boundary of pancreatic islet toward the ACE lumen. Faint green (excitation 458 nm wavelength) and red (excitation 561 nm wavelength) signals are autofluorescence. CTLs marked by GzmB-mTFP (green, excitation 488 nm) infiltrated the islet during the immune response. (**D**) Velocity of 81 migrating CTLs tracked from panel (**C**) after imaging for >30 min. Mean is shown as a plus symbol, median as a line. See also *Figure 5—video 1*. (**E**) Two-photon image of GzmB-mTFP positive cells from the corneal area. Individual vesicles inside the CTL are easily identified (inset). See also *Figure 5—video 2*.

The online version of this article includes the following video and figure supplement(s) for figure 5:

**Figure supplement 1.** Longitudinal and repetitive observations of transplanted islets inside the ACE.

**Figure 5—video 1.** In vivo imaging of migrating T cells during infiltration of a transplanted pancreatic islet.
https://elifesciences.org/articles/58065#fig5video1

**Figure 5—video 2.** In vivo imaging of migrating T cells in the cornea by two-photon microscopy Same experiment as in *Figure 5E*.
https://elifesciences.org/articles/58065#fig5video2

specific antibodies against CD4+ T helper cells and CD8+ CTLs (*Figure 5B*). We found almost no overlap between GzmB-mTFP positive cells and CD4+ cells, while the majority of CD8+ cells also were GzmB-mTFP positive. These data again demonstrate the high specificity of the GzmB-mTFP-KI as a marker for CTLs. The small number of GzmB-mTFP positive, but CD8 negative cells might represent natural killer (NK) cells.

Next, we used the ACE model for long-term in vivo observations of CTL migration. We imaged islets in 40 µm thick layers (11 planes) and analyzed the migration of invading CTLs in 4D (volume and time) as previously described (*Abdulreda et al., 2011*). *Figure 5C* shows a representative, confocal overview of an islet with green spots indicating the positions of single CTLs. *Figure 5—video 1* shows the respective long-term observation over 32 min. Analysis of CTL velocity from these data

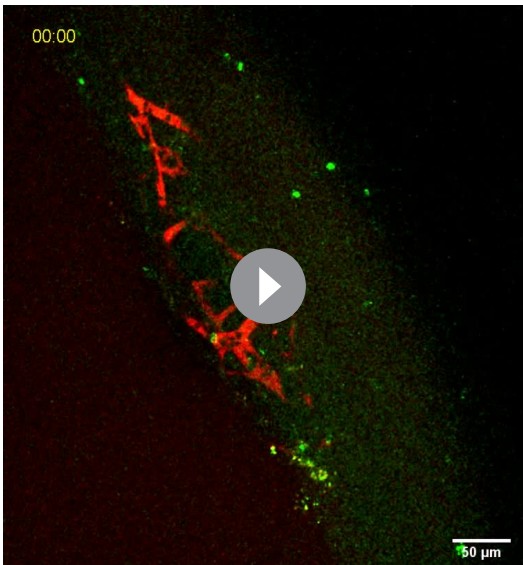

**Video 1.** In vivo imaging of the capillary network of a transplanted pancreatic islet. The pancreatic islet was transplanted into the ACE 14 days prior the experiment. The movie is a maximum intensity projection of 11 planes covering a 40 µm thick stack over 37 min and was acquired with a two-photon microscope tuned to 880 nm for GzmB-mTFP (CTL marker) and 1040 nm for rhodamine-dextran (MW 70 kD; blood vessel marker).
https://elifesciences.org/articles/58065#video1

yielded an average migration velocity of 5.52 ± 1.5 µm/min (mean ±s.d.; median 5.34) at POD14 (*Figure 5D*). At later PODs, the velocity increased due to the advanced tissue clearance (data not shown).

Ultimately, the specific, endogenous labeling of individual granules should enable us to visualize single fusion events of CG in vivo. A prerequisite for this goal is to achieve sufficient resolution and acquisition frequencies to discriminate single vesicles and track them. As shown in *Figure 5— videos 1* and *2* and in *Figure 5E*, we clearly identified individual granules in migrating CTLs. The nominal lateral resolution in the raw data of the videos and the image in *Figure 5E* was ~0.7 µm (pixel size 340 × 340 nm). The apparent resolution as determined by 'full width at half maximum (FWHM)' analysis was dependent on the recording conditions and the position of the cell within an islet. Typically within the first 40 µm from the islet surface, we reached a resolution of ~1 µm. While using two-photon instead of confocal microscopy (*Figure 5E*) did not increase the maximal resolution, the signal to noise ratio and the light-penetration depth were substantially increased. New technological developments, like a combination of two-photon microscopy with STED microscopy (*Takasaki and Sabatini, 2014*) will further improve 3D resolution in in vivo studies and will, in combination with the GzmB-mTFP-KI, allow the observation of individual CG fusion events in vivo.

## Discussion

The elimination of infected or malignant cells through CTL-mediated target cell killing is essential to maintain health. While in vitro studies indicated that CTL-mediated target cell killing is highly efficient, two-photon imaging of lymph nodes in living mice demonstrated that CTLs kill on average only 4.5 target cells per day and require the cooperativity of several CTLs acting in concert (*Halle et al., 2016*; *Nolz and Hill, 2016*; *Halle et al., 2017*). This surprising inefficiency in vivo is most likely caused by rate-limiting steps like CTL priming, CTL trafficking, CTL properties and target cell properties (*Halle et al., 2017*). For the optimization of vaccines and CTL-based immunotherapies it is therefore of utmost importance to study CTL behavior in the physiological setting of a living organism.

We report the generation of a GzmB-mTFP-KI, in which all CGs of T cells are endogenously fluorescent due to an mTFP-tag at the CG-resident enzyme GzmB. We chose the mTFP tag because of its ideal suitability for two-photon imaging in vivo and for Förster resonance energy transfer (FRET) (*Day et al., 2008*; *Drobizhev et al., 2011*; *Gossa et al., 2015*). mTFP is a derivative of cyan fluorescent protein (CFP) from *Clavularia* coral (*Ai et al., 2006*), is monomeric (26.9 kDa) unlike the tetrameric CFP, has a high brightness, photostability and quantum yield, is only weakly acid sensitive and has a narrow emission spectrum (*Ai et al., 2006*; *Day et al., 2008*). The brightness, photostability and high quantum yield of mTFP are ideally suited for longitudinal, non-invasive imaging over extended time periods. In addition, mTFP is tolerated by the immune system upon adoptive transfer, which favors its use for imaging immune cells in vivo (*Gossa et al., 2015*).

The biggest concern when using fusion protein tags is that the attached tag might interfere with the expression, localization or function of the tagged protein, in our case GzmB. GzmB is

synthesized as a zymogen and cleaved by cathepsin C in lysosomes (*Jenne and Tschopp, 1988*; *Krahenbuhl et al., 1988*; *Caputo et al., 1993*; *Perišić Nanut et al., 2014*; *Voskoboinik et al., 2015*), which removes an N-terminal 18-residue signal peptide and the downstream activation dipeptide Gly-Glu (*Caputo et al., 1993*). To avoid interference with proper GzmB localization we therefore chose the C-terminus for fusion of mTFP. Further, we added a flexible GGSGGSGGS linker between GzmB and mTFP, which is cleaved in CGs (*Figure 1C*; *Huang et al., 2014*). Our co-localization analyses with perforin and the CG v-SNARE synaptobrevin2 (*Figure 2*) and TIRF imaging of individual CG fusion events (*Figure 3*) demonstrate that the mTFP and GzmB in the GzmB-mTFP-KI are correctly and specifically localized to CG, so that GzmB is not functionally compromised (see below) and mTFP can be used as a very faithful CG marker.

Beyond subcellular protein trafficking and localization, tags can affect protein function. Indeed, a recently published GzmB-tdTomato-KI exhibits reductions of 75% in GzmB expression, of 30% in CTL degranulation and of 25% in GzmB protease activity (*Mouchacca et al., 2013*; *Mouchacca et al., 2015*). In striking contrast to this, our analyses of GzmB-mTFP-KI CTLs did not reveal any changes in GzmB expression (*Figure 1C–D*), CG transport (*Figure 2G–H*), CG fusion efficiency (*Figure 3C–D*), degranulation capacity (*Figure 4B*) or killing efficiency (*Figure 4A*). This may be partly due to the smaller size of mTFP as compared to tdTomato (26.9 vs. 54.2 kDa). More importantly, though, the fact that GzmB-mTFP appears to be efficiently cleaved into GzmB and mTFP in the acidic environment of the CG lumen (*Figure 1C–D*) contributes substantially to the almost perfect functionality of GzmB in the GzmB-mTFP-KI, so that GzmB can operate in an unperturbed manner while mTFP can still be used to follow CG localization, movement and fusion.

The advantages and potential of the novel GzmB-mTFP-KI are manifold. While many current in vivo studies require the removal of CTLs from lymphatic tissue and their exogenous labeling and re-injection via the tail vein, no manipulations are necessary for confocal or two-photon imaging in the GzmB-mTFP-KI (*Figure 5*). Further, the subcellular targeting of GzmB-mTFP to CGs allows the investigation of individual CGs without compromising analyses of single CTLs in tissue. In combination with FRET-based apoptosis reporters (*Breart et al., 2008*; *Shcherbo et al., 2009*; *Garrod et al., 2012*; *Cazaux et al., 2019*) expressed in the islets the GzmB-mTFP-KI might shed new light on the degranulation process in vivo. In conclusion, the new GzmB-mTFP-KI mouse now allows the investigation of rate-limiting processes in CTL function like priming, differentiation, migration and killing in order to optimize vaccines and immunotherapies for virus infections and cancer.

# Materials and methods

## Key resources table

| Reagent type (species) or resource | Designation | Source or reference | Identifiers | Additional information |
|---|---|---|---|---|
| Antibody | anti-GAPDH (rabbit polyclonal) | Cell Signaling Technology | Cat# 2118L; Clone-14C10 | WB (1:5000) |
| Antibody | anti-tRFP (rabbit polyclonal) | Evrogen | Cat# ab233 | WB (1:2000) |
| Antibody | anti-Granzyme B (rabbit polyclonal) | Cell Signaling Technology | Cat# 4275S | WB (1:2000) |
| Antibody | Alexa647-coupled anti-Granzyme B (mouse IgG1, κ) | BioLegend | Cat# 515405; clone-GB11 | IF (1:200) |
| Antibody | Alexa647-coupled anti-perforin (mouse, IgG2b, κ) | BioLegend | Cat# 308109; clone-dG9 | IF (1:200) |
| Antibody | Hamster anti-mouse (CD3ε) | BD Pharmingen | Cat# 553058; clone-145–2 C11 | 30 µg/ml |
| Antibody | Rat Brilliant Violet421 anti-mouse CD8a (rat, IgG2a, κ) | BioLegend | Cat# 100737; clone-53–6.7 | IHC (1:200) |

*Continued on next page*

*Continued*

| Reagent type (species) or resource | Designation | Source or reference | Identifiers | Additional information |
|---|---|---|---|---|
| Antibody | Rat APC anti-mouse CD4 (rat, IgG2b, κ) | BioLegend | Cat# 100412; clone-GK1.5 | IHC (1:200) |
| Antibody | Rat anti-mouse CD107a-PE (rat, IgG2a, κ) | BioLegend | Cat# 121611; clone-ID4B | FACS (1:200) |
| Recombinant DNA reagent | Casper3-GR (plasmid) | Evrogen | Cat# FP971 | |
| Recombinant DNA reagent | pMAX-Granzyme B-mCherry (mouse) | This paper | | *Figure 3*, Available from the authors upon request |
| Recombinant DNA reagent | pMAX-Casper3-GR (plasmid) | This paper | | *Figure 4A*; Available from the authors upon request |
| Biological sample (*Mus musculus*) | Primary CD8+ T lymphocytes | This paper | | Freshly isolated from spleen of Granzyme B-mTFP knock-in mouse (8–20 weeks old); Mouse available from EMMA mouse depository |

## Generation of GzmB-mTFP KI mice and other animals used

GzmB-mTFP-KI mice were generated by CRISPR-Cas9 technology with an HDR fragment designed to replace the endogenous Stop codon of the *Gzmb* gene by sequences encoding a GGSGGSGGS-linker, mTFP and a Stop codon (*Figure 1A*, *Figure 1—figure supplement 1*). Mouse *Gzmb* gene sequences required for the HDR fragment were PCR amplified from genomic C57BL/6J (Janvier) DNA, subcloned into pMiniT (NEB) and sequence verified, and the sequence encoding GGSGGSGGS-mTFP-Stop was obtained from a previously generated plasmid (pMAX-Synaptobrevin2-mTFP). Using these sequences, the HDR fragment was generated by Gibson assembly (*Gibson, 2009*). For zygote injections, we used a single sgRNA (5'-GTC CAG GAT TGC TCT AGG AC-3'; PAM, AGG), in vitro transcribed capped Cas9-D10A mRNA ('nickase' version of Cas9 [*Gasiunas et al., 2012*; *Jinek et al., 2012*; *Cong et al., 2013*]), and the purified HDR fragment. The strategy to use a single sgRNA in combination with Cas9-D10A mRNA was chosen to minimize the likelihood of off-target cleavage events. The CRISPR-Cas9 reagents (10 ng/µl sgRNA, 10 ng/µl Cas9-D10A mRNA, 20 ng/µl HDR fragment) were injected into the pronucleus and cytosol of zygotes (C57BL/6J, Janvier), which were then re-implanted into seven recipient mice. We obtained 32 offspring, which were genotyped for the proper insertion of the HDR fragment by PCR, using primer 36016 (ATCAAAGAACAGGAGAAGACCCAG, Exon 3) in combination with primer 33615 (GGTG TTGGTGCCGTCGTAGGG, mTFP) (1433 bp fragment) and primer 19524 (ACCGCATCGAGATCC TGAACC, mTFP) in combination with primer 36017 (AATGGCTAAGCAATCCCATCAGG, downstream of HDR1) (1565 bp fragment). Of the 32 offspring, one carried the desired mutation. This mouse was subsequently bred to C57BL/6J mice (Janvier) to obtain germline transmission of the GzmB-mTFP-KI mutation. Genotyping of later generations was done by short-range PCR (*Figure 1A*, *Figure 1—figure supplement 1*). As soon as the paper is published, the GzmB-mTFP-KI mouse line will be deposited at the European Mouse Mutant Archive (EMMA). Until the line is established at EMMA, mice will be provided by the authors upon request. C57BL/6 mice for other experiments were purchased from Charles River Laboratories (Sulzfeld, Germany). Donor mice for islet of Langerhans isolation were DBA/2 mice (12–30 weeks old). All animal experiments were performed according to German law and European directives, and with permission of the Niedersächsisches Landesamt für Verbraucherschutz und Lebensmittelsicherheit (LAVES animal license number 33.9-42502-04-13/1359) and the state of Saarland (Landesamt für Gesundheit und Verbraucherschutz; animal license number 41–2016).

## Plasmids and antibodies

Casper3-GR (FP971; Evrogen) vector was digested with BamH1 and Not1 to remove TagRFP-linker-TagGFP. The same digestion was performed for pMAX vector and both vector and insert were ligated. The following primary antibodies were used: anti-GAPDH (RRID:AB_561053), anti-tRFP (RRID:AB_2571743) and anti-GzmB (RRID:AB_2114432). Secondary antibodies were HRP-conjugated donkey anti-rabbit and Alexa Fluor 647 goat anti-mouse IgG (H+L) (Thermo Fisher Scientific). For structured illumination microscopy (SIM), Alexa647-coupled anti-granzymeB (RRID:AB_2294995) and Alexa647-coupled anti-perforin (RRID:AB_493255) antibodies were used. For total internal reflection fluorescence microscopy (TIRFM) a hamster anti-mouse CD3ε (RRID:AB_394591) antibody was used for coating coverslips and stimulating cells. For FACS, a rat anti-mouse CD107a-PE (LAMP-1-PE) (RRID:AB_1732051) antibody was used.

## Cell culture

Splenocytes were isolated from 8 to 18 week-old C57BL6/N, Syb2-mRFP/GzmB-mTFP double knock-in mice or Granzyme B-mTFP-KI mice, as described before (Chang et al., 2016). Briefly, naive CD8$^+$ T cells were isolated from splenocytes using Dynabeads FlowComp Mouse CD8$^+$ kit (Thermo Fisher Scientific). The isolated naive CD8$^+$ T cells were cultured for up to 12 to 14 d (37°C, 5% CO$_2$) with anti-CD3/anti-CD28 activator beads with 100 U/mL IL2 (1:0.8 ratio) at a density of $1 \times 10^6$ cells/mL in AIMV (Thermo Fisher Scientific) containing 10% FCS and 50 µM 2-ME to generate effector CTLs. P815 target cells were cultured in RPMI 1640 medium (Thermo Fisher Scientific) containing 10% FCS, 1% pen/strep, and 10 mM HEPES.

## CTL:target cell conjugation

For functional effector:target cell conjugations P815 target cells were incubated with 10 µg/ml anti-CD3ε antibody for 15–30 min at 37°C and washed with 1xPBS. $0.2 \times 10^6$ P815 cells were plated on 12 mm glass coverslips. Carefully, $0.2 \times 10^6$ CTLs were added on top of settled P815 target cells and imaged for 45 min to 1 hr.

## Structured illumination microscopy (SIM)

Cells were fixed in ice-cold 4% PFA in Dulbecco's 1xPBS (Thermo Fisher Scientific) for 20 min. For staining, cells were permeabilized with 0.1% TritonX-100 in Dulbecco's 1xPBS (permeabilizing solution) for another 20 min followed by 30 min blocking in solution containing 2% BSA prepared in permeabilizing solution. Cells were stained with either Alexa 647 conjugated anti-GzmB or Alexa 647 conjugated anti-perforin antibodies (BD Biosciences) and observed at an Elyra PS.1 microscope (Zeiss, Jena, Germany). Images were acquired with a 63x Plan-Apochromat (NA 1.4) with laser excitation at 488, 561, and 635 nm and then processed to obtain higher resolutions (RRID:SCR_013672). For analysis of co-localization, Pearson's and Manders' coefficients of correlation (Pearson, 1909; Manders et al., 1993) were determined using the JACoP plugin of ImageJ v1.46.

## Total internal reflection fluorescence (TIRF) microscopy

The TIRFM setup (Visitron Systems GmbH, Puchheim, Germany) was based on an IX83 (Olympus) equipped with the Olympus autofocus module, a UAPON100XOTIRF NA 1.49 objective (Olympus), a 488 nm 100 mW laser and a solid-state laser 100 mW emitting at 561 nm, the iLAS2 illumination control system (Roper Scientific SAS, France), the evolve-EM 515 camera (Photometrics) and a filter cube containing Semrock (Rochester, NY, USA) FF444/520/590/Di01 dichroic and FF01-465/537/623 emission filter. The setup was controlled by Visiview software (version 4.0.0.11, Visitron GmbH). For TIRFM, day seven bead activated CTLs isolated from GzmB-mTFP-KI or GzmB-mTFP/Syb2-mRFP double knock-in mice were used. 2 hr before experiment, beads were removed from CTLs and roughly (0.2–0.3 $\times 10^6$ cells) were resuspended in 30 µL of extracellular buffer (2 mM HEPES, 140 mM NaCl, 4.5 mM KCl, and 2 mM MgCl$_2$) containing no Ca$^{2+}$ and allowed to settle for 1–2 min on anti-CD3ε antibody (30 µg/mL) coated coverslips. Cells were then perfused with extracellular buffer containing 10 mM Ca$^{2+}$ for visualizing CG fusion. Cells were imaged for 7 min at room temperature at 488 nm or alternating between 488 nm and 561 nm with acquisition frequency of 10 Hz and exposure time of 100 ms. Images and time-lapse series were analyzed using ImageJ (RRID:SCR_003070) or the FIJI package of ImageJ (RRID:SCR_002285). CG fusion analysis was performed using ImageJ

with the plugin Time Series Analyzer (RRID:SCR_014269). A sudden drop in GzmB-mTFP or GzmB-mCherry fluorescence occurring within 300 ms (three acquisition frames) was defined as fusion (Ming et al., 2015). The number of vesicles was shown according to the corresponding fluorescence intensity of the vesicles.

## Western blot analysis

Mouse CTLs were homogenized with a syringe in lysis buffer (50 mM Tris (pH 7.4), 1 mM EDTA, 1% Triton X-100, 150 mM NaCl, 1 mM DTT, 1 mM deoxycholate, protease inhibitors, and PhosSTOP; Roche) on ice and insoluble material was removed by centrifugation at 11,300 g for 10 min. The protein concentration was determined using Quick Start Bradford 1x Dye Reagent (5000205; Bio-Rad). Proteins were separated by SDS-PAGE (NuPAGE; Thermo Fisher Scientific), transferred to nitrocellulose membranes (Amersham), and blocked by incubation with 5% skim milk powder in 20 mM Tris, 0.15 M NaCl, pH 7.4, and TBS for 1–2 hr and blotted with specific antibodies. Blots were developed using enhanced chemiluminescence reagents (SuperSignal West Dura Chemoluminescent Substrate; Thermo Fisher Scientific) and scanned.

## Degranulation assay

CTLs isolated from wt and GzmB-mTFP-KI mouse were stimulated with anti-CD3/CD28 beads. Day 3, 5, 7 and 10 activated CTLs from both wt and GzmB-mTFP knock-in mouse were used for degranulation assay. 2 hr before degranulation, anti-CD3/CD28 beads were removed from cells and allowed the cells to rest in the 37°C incubator. $0.2 \times 10^6$ CTLs were used for each condition. Cells were plated in duplicates in 96-well plates either coated with or without anti-CD3/CD28 Ab, along with antibiotic Monensin (BD Biosciences) to block endocytosis of CD107a (LAMP1a) from the CTL membrane. Cells were allowed to degranulate for 2 hr at 37°C. Cell were washed and resuspended in isolation buffer (0.1% BSA + 2 mM EDTA in D-PBS, pH 7.4) for acquisition. Degranulation was measured with anti-CD107a conjugated with PE (RRID:AB_1732051). Data were acquired in FACSAria3 (BD Pharmingen). Data were analyzed using FlowJo software (RRID:SCR_008520). Gates were set based on unstained controls.

## Live-cell killing assay using Casper3-GR target cells

Briefly, $0.2 \times 10^4$ P815 target cells stably expressing Caspr3-GR (FRET construct containing Tag-GFP and Tag-RFP with a target cleavage site DEVD of Caspase 3 (activated via GzmB) were washed once with PBS and resuspended in 100 μl AIMV medium. P815 cells were incubated with 30 μg/ml anti-CD3 antibody at 37°C for 20 min and plated into 8-well strips (BD Falcon). After 2–3 min, $0.2 \times 10^6$ day 5 CTLs from either wild type or granzyme B-mTFP knock-in mouse were added gently to the target cells and imaged for 45 min to 1 hr. The cytotoxicity was calculated from the decrease in Tag-RFP and increase in Tag-GFP fluorescence. Fold ratio change of Tag-GFP/Tag-RFP values were calculated using ImageJ (RRID:SCR_003070) software.

## Isolation of islets of Langerhans

Islets of Langerhans were isolated as described (Stull et al., 2012). In short, mice were killed by cervical dislocation. After opening of the abdominal cavity, the bile duct was ligated by use of surgical suture near to the liver. A cut (0.5 mm) was made into the intestine where the bile duct ended (papilla). Through that cut a 0.40 mm wide blunt cannula was inserted into the bile duct half way toward the ligation. Through that cannula 3 mL of a Ringer solution containing 50 μg Liberase TL (Merck) was slowly injected into the bile duct, and inflated the pancreatic tissue. After injection, the pancreas was removed from the mouse abdomen. For digestion the pancreas was heated up to 37°C in 5 mL of Ringer solution and was gently agitated in a prewarmed water bath for 12 min. Digestion was stopped by adding 25 mL of cold (4°C) Ringer solution. The pancreas then was smashed mechanically. Undigested debris was separated from Islets and exocrine material by filtration through a steel mesh (200 μm mesh size). Finally islets were separated from exocrine material by a density centrifugation (900 g, 18 min) using a Ringer/Histopaque gradient (Histopaque 1077, Merck). Islets are supposed to stay at the Ringer/Histopaque interphase, while exocrine material goes to the pellet. After isolation islets were collected by filtering the supernatant through a 70 μm strainer,

washed and resuspended in RPMI 1740 cell culture medium and kept in an incubator (37°C, 5% $CO_2$) for up to 4 days before further use.

## Transplantation of islets of Langerhans into the anterior chamber of the eye

Transplantation was performed as described (*Abdulreda et al., 2013*). In short, mice (12–20 weeks old) were anesthetized with isoflurane. An opening (500 µm) was cut into the cornea with a sharp injection needle (30 G). Through this opening a blunt cannula (0.4 mm) preloaded with 10–20 islets was inserted. The cannula was connected to a foot panel-steered injector through which the islets were injected into the eye under optical control (binocular microscope). After transplantation mice were kept for 4 days under pain relief medication (Caprofen, 5 mg/kg bodyweight) before any further experimental manipulation.

## Microscopy of animals transplanted with intraocular islets

Imaging was performed as described previously (*Abdulreda et al., 2011*). In short, transplanted mice were pre-anesthetized with isoflurane in a box. Under anesthetization animals were then mounted to a three point head holder with a respirator mask for continuing anesthesia. The mouse laid on a heating plate and the head was tilted in a way that the corneal surface of the transplanted eye was in perpendicular orientation to an objective of an upright microscope. Two different microscopes (LSM880 confocal microscope, LSM880 two-photon microscope, both Carl Zeiss Jena) were used for live imaging. Both systems were equipped with a 20x Plan-Apochromat water immersion objective (Carl Zeiss Jena, NA 1.0). If not otherwise stated data were acquired as volume (xyz) and time (t) 4D stacks (xyzt; 11 planes over 40 µm, 30–60 s interval) and are displayed as maximum intensity projections. For data acquisition Zen software (RRID:SCR_013672) was used. Data analyses and 3D reconstructions were performed by using Imaris9.3 (RRID:SCR_007370). Deconvolution was done with Autoquant X3 (RRID:SCR_002465).

## Immunhistochemistry

For immunhistochemistry mice were deeply anesthetized by a intraperitoneal injection of a mixture of ketamine (280 mg/kg bodyweight) and xylacine (20 mg/kg body weight). Access to the heart was obtained by removing part of the rips together with the sternum and lung displacement. Subsequently an injection needle connected to a perfusion pump punctured the left ventricle, and the right atrium was opened with small scissors. Perfusion was started immediately by a perfusion pump. Approximately 200 mL of fixative (Ringer solution + 4% paraformaldehyde) was pumped through the vascular network of the mouse.

After perfusion the transplanted eye was isolated and postfixed in the same fixative for another 4 hr. After washing (3x, 2 hr each) the eye was incubated in 30% sucrose solution over night (4°C) before embedding in Tissue-Tek O.C.T. compound (Sakura Finetek Europe B.V.). For embedding the eye was placed in a small beaker and surrounded by Tissue-Tek O.C.T. compound, dipped for 1 min into 2-methylbutane at −80°C and stored at −80°C until cutting. A cryostat (Thermo Scientific HM525) was used to produce 10–20 µm thick eye slices. Slices were collected on Superfrost+ slides (VWR). Staining with primary and secondary antibodies followed standard protocols. Samples were mounted using ProLong Gold Antifade (RRID:SCR_015961) as a mounting medium.

## Statistics and image analyses

Statistical differences in data were calculated with paired or unpaired Student's *t*-test. Unless mentioned otherwise, all data are presented as average ± standard average of the mean (s.e.m.). Data were analyzed with ImageJ v1.46 (RRID:SCR_003070) (*Schneider et al., 2012*), Excel (RRID:SCR_016137), SigmaPlot 13 (RRID:SCR_003210) and Imaris 9.3 (RRID:SCR_007370) and graphed using Affinity Designer Software (RRID:SCR_016952).

## Acknowledgements

This work was supported by grants from the Deutsche Forschungsgemeinschaft (SFB 894 (TL-Z and JR), INST 256/427–1 FUGB (TL-Z) and IRTG 1830 (JR). We thank Margarete Klose, Anja Bergsträßer,

Tamara Brück, Nicole Rothgerber and Katrin Sandmeier for excellent technical assistance, and Ursula Fünfschilling and Monika Schindler of the MPIEM Transgene Facility for mouse embryology and zygote injections.

## Additional information

### Competing interests

Nils Brose: Reviewing editor, *eLife*. Midhat H Abdulreda: works as a consultant for Biocrine, an unlisted biotech company that is using the ACE technique as a research tool. Per-Olof Berggren: is cofounder and CEO of Biocrine, an unlisted biotech company that is using the ACE technique as a research tool. The other authors declare that no competing interests exist.

### Funding

| Funder | Grant reference number | Author |
| --- | --- | --- |
| Deutsche Forschungsgemeinschaft | SFB894/A10 | Jens Rettig |
| Deutsche Forschungsgemeinschaft | SFB894/P1 | Elmar Krause<br>Jens Rettig |
| Deutsche Forschungsgemeinschaft | IRTG1830 | Keerthana Ravichandran<br>Jens Rettig |
| Deutsche Forschungsgemeinschaft | SFB894/A17 | Trese Leinders-Zufall |

The funders had no role in study design, data collection and interpretation, or the decision to submit the work for publication.

### Author contributions

Praneeth Chitirala, Paloma Martzloff, Keerthana Ravichandran, Investigation; Hsin-Fang Chang, Claudia Schirra, Investigation, Methodology; Christiane Harenberg, Per-Olof Berggren, Trese Leinders-Zufall, Fritz Benseler, Resources, Methodology; Midhat H Abdulreda, Methodology; Elmar Krause, Data curation, Formal analysis, Methodology; Nils Brose, Resources, Supervision, Funding acquisition, Methodology, Writing - original draft, Writing - review and editing; Jens Rettig, Conceptualization, Supervision, Funding acquisition, Validation, Writing - original draft, Writing - review and editing

### Author ORCIDs

Praneeth Chitirala https://orcid.org/0000-0002-5540-9300
Hsin-Fang Chang https://orcid.org/0000-0002-7691-4090
Elmar Krause http://orcid.org/0000-0003-0891-2373
Trese Leinders-Zufall http://orcid.org/0000-0002-0678-362X
Jens Rettig https://orcid.org/0000-0001-6160-3954

### Ethics

Animal experimentation: All animal experiments were performed according to German law and European directives, and with permission of the Niedersächsisches Landesamt für Verbraucherschutz und Lebensmittelsicherheit (LAVES animal license number 33.9-42502-04-13/1359) and the state of Saarland (Landesamt für Gesundheit und Verbraucherschutz; animal license number 41-2016).

### Decision letter and Author response

Decision letter https://doi.org/10.7554/eLife.58065.sa1
Author response https://doi.org/10.7554/eLife.58065.sa2

## Additional files

### Supplementary files

- Source data 1. Chitiralaetal_Source Data.xlsx.

- Transparent reporting form

### Data availability

As soon as the paper is published, the GzmB-mTFP KI mouse line will be deposited at the European Mouse Mutant Archive (EMMA). Until the line is established at EMMA, mice will be provided by the authors upon request. A source data file has been provided for Figures 1-4.

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
