## [Decision Letter]

**Acceptance summary:**

This manuscript describes a new and interesting mouse model to study CTL (and probably NK cell) activity in vivo by tracking Granzyme B levels and localisation in real-time. This will be an excellent resource for groups studying a range of T and NK cell functions in many contexts from anti-viral defences to tumour immunotherapy.

**Decision letter after peer review:**

Thank you for submitting your article "Studying the biology of cytotoxic T lymphocytes in vivo with a fluorescent Granzyme B-mTFP knock-in mouse" for consideration by *eLife*. Your article has been reviewed by three peer reviewers, including Michael L Dustin as the Reviewing Editor and Reviewer #1, and the evaluation has been overseen by Satyajit Rath as the Senior Editor. The following individual involved in review of your submission has agreed to reveal their identity: Jennifer L Stow (Reviewer #2).

The reviewers have discussed the reviews with one another and the Reviewing Editor has drafted this decision to help you prepare a revised submission.

Summary:

Advanced in vivo microscopy offers the ability to track fluorescently tagged cells, organelles and proteins in their native tissue environments providing invaluable insights for physiology and pathophysiology. Moreover, the ability to do so seamlessly, without perturbing endogenous localization or function, is the ultimate goal for such studies. The behaviour of cytotoxic granules (CGs) in T cells has long been a preoccupation of immunologists and others, due to their essential role in cell-cell killing, now a topic of intense interest in cancer immunotherapies.

Your study describes the construction of a fluorescent fusion protein GzmB- mTFP consisting of mTFP, a cleavable linker and the CG enzyme, granzymeB and the generation of a knock-in mouse expressing this probe in CTLs. Elegant imaging is used to show remarkable localization and colocalization of the fusion protein and other markers in CGs. The behaviour of the fusion protein and CGs are carefully measured in ex vivo cells and in mice, to ensure no aberrant features have been introduced. With no impediment to normal function, this CG probe appears to be superior to others available. Finally, the WT and KI mice are subjected to allografts in the eye to provide a very nice demonstration in vivo of cell infiltration and granule formation in CTLs. More sophisticated studies in the future no doubt will allow discrimination of new aspects of CG behaviour in cell-cell killing using further super-resolution microscopy and CLEM.

You are commended on a rigorously performed study, the KI mouse and its protein GzmB-mTFP cells will be extremely valuable for many investigators and the mice have the hallmarks of becoming iconic models for future CG studies. As such, this study represents a major, enabling contribution to the field. There are some issues that should be addressed. First of all, as this will be a resource for the field, how will you make the mice available to the community?

If the data are available some quantification of granzyme and fusion protein levels in Figures 1 C and D would be helpful since the relative levels in WT and KI mice appear to differ between these panels. It would certainly be interesting in the future to follow the expression of the fusion protein over longer time periods and under different stimulation conditions in the KI mice.

What is the expression of the reporter in NK cells and activated CD4 T cells? A more detailed description of the model would be welcome. In the in vivo experiment, it would be important that the author stained for CD8 T cell and NK cells. A negative result on a CD4 staining is not enough to claim that cells are CD8 T cells. For the in vivo imaging in general it would be important to show controls with T1D and similar/identical imaging conditions in the micro without the knockin. The green fluorescence channel has a lot of autofluorescence and that autofluorescence might appear granular, so it would be helpful to see evidence that this is specific mTFP fluorescence.

Revisions expected in follow-up work:

Do you have a way to look at the kinetics of the cleavage process, perhaps through a pulse-chase experiment or cruder kinetic study of granule formation during CTL maturation.

In this context, could the authors comment on whether they see any evidence for cleavage of the fusion protein before the CG (eg at the level of post-Golgi vesicles) where proteolytic processing is possible? Can cleavage be prevented in the lysosome/granules after dissipating the pH drop, or by incubation of the cells with protease inhibitors that would intersect at the endo-lyosomal comparments? This is not necessary to do immediately but of future interest, but if you have data on this it would strengthen it.

Can you draw new conclusions from the longitudinal analyse of the T1D model? New biology is not needed at this stage, but it should be clear that the tools will be useful for generating quantitative data to address open questions in biology.

---

## [Author Response]

Revisions for this paper:You are commended on a rigorously performed study, the KI mouse and its protein GzmB-mTFP cells will be extremely valuable for many investigators and the mice have the hallmarks of becoming iconic models for future CG studies. As such, this study represents a major, enabling contribution to the field. There are some issues that should be addressed. First of all, as this will be a resource for the field, how will you make the mice available to the community?

As soon as the paper is published, we will deposit the line at the European Mouse Mutant Archive (EMMA). Until the line is established at EMMA, mice will be provided by the authors upon request. We have now stated this in the Materials and methods section of the revised manuscript.

If the data are available some quantification of granzyme and fusion protein levels in Figures 1 C and D would be helpful since the relative levels in WT and KI mice appear to differ between these panels. It would certainly be interesting in the future to follow the expression of the fusion protein over longer time periods and under different stimulation conditions in the KI mice.

It is true that the expression levels of GzmB differ between each preparation, also in WT. We have quantified the levels on the blots shown in Figure 1C and D and give the values in the Results section (paragraph one). We fully agree that the expression kinetics of the fusion protein, also in comparison to the corresponding WT levels, should be done quantitatively in the future. We can state, however, that in the imaging experiments shown in Figures 2-5 we always observed a robust fluorescence without the requirement to change the intensity of the excitation laser(s).

What is the expression of the reporter in NK cells and activated CD4 T cells? A more detailed description of the model would be welcome. In the in vivo experiment, it would be important that the author stained for CD8 T cell and NK cells. A negative result on a CD4 staining is not enough to claim that cells are CD8 T cells. For the in vivo imaging in general it would be important to show controls with T1D and similar/identical imaging conditions in the micro without the knockin. The green fluorescence channel has a lot of autofluorescence and that autofluorescence might appear granular, so it would be helpful to see evidence that this is specific mTFP fluorescence.

We have tested the expression of the reporter in CD4^+^ and CD8^+^ T cells by antibody staining in the anterior eye chamber. Based on these stainings there is little to none expression of GzmB in CD4^+^ T cells infiltrating the islet at POD14. We do find, however, a small fraction of GzmB-mTFP^+^ cells which are CD4^-^CD8^-^. These cells most likely represent NK cells, for which we do not currently have a specific staining. We have revised the Figure 5 and now show the CD4/CD8 stainings in panel B.

Revisions expected in follow-up work:Do you have a way to look at the kinetics of the cleavage process, perhaps through a pulse-chase experiment or cruder kinetic study of granule formation during CTL maturation.

A pulse-chase experiment does not work in this case as the mutant is not inducible. One could do sub-cellular fractionation, purify different granule/organelle populations and quantify the amount of the GzmB-mTFP fusion protein and the cleavage products GzmB and mTFP. We will initiate these experiments after the lock-down, also in an effort to purify fusogenic granules based on their fluorescence.

In this context, could the authors comment on whether they see any evidence for cleavage of the fusion protein before the CG (eg at the level of post-Golgi vesicles) where proteolytic processing is possible? Can cleavage be prevented in the lysosome/granules after dissipating the pH drop, or by incubation of the cells with protease inhibitors that would intersect at the endo-lyosomal comparments? This is not necessary to do immediately but of future interest, but if you have data on this it would strengthen it.

We agree that it would be useful to know whether proteolytic cleavage of the GzmB-mTFP fusion protein could be prevented at earlier CG maturation steps. We envision the dissipation of the pH drop to be difficult, because we have recently demonstrated that this manipulation leads to a change in diameter, density and microtubular transport of CGs (PMID: 32269094). The treatment with protease inhibitors might be an attractive alternative which could be tested in the future.

Can you draw new conclusions from the longitudinal analyse of the T1D model? New biology is not needed at this stage, but it should be clear that the tools will be useful for generating quantitative data to address open questions in biology.

We indeed developed the GzmB-mTFP knock-in mouse in combination with the ACE to generate quantitative data addressing open questions in biology. Our interest is in the function of immune cells in vivo, which is why the focus will be on immune diseases like FHL rather than on diabetes like in T1D. We have initiated these experiments before the lock-down by crossing the GzmB-mTFP knock-out with perforin knock-out mice, which serves as a model for familial hemophagocytic lymphohistiocytosis (FHL) type 2. As evident from the attached image we confirmed in in vitro experiments that perforin-deficient CTLs have reduced killing capacity. In upcoming in vivo experiments in the ACE we now want to investigate whether these CTLs also have additional deficits (e.g. in migration) and whether the remaining killing ability is mediated by the Fas/FasL pathway. We just started the breeding of the double-transgenic mice again in our animal facility and expect the first offspring in about six months.

**Author response image 1. sa2fig1:** Islets of Langerhans were incubated with CTLs derived from GzmB-mTFP KI (left) and GzmB-mTFP KI/Perforin KO mice (middle) and imaged by confocal microscopy. CTLs started to infiltrate the islets and kill them, which resulted in a detachment of islets from the culture dish. Quantification of the detachment at day 1 and day 2 of co-culture (right panel) shows a clear killing deficit in GzmB-mTFP KI/Perforin KO compared to GzmB-mTFP KI alone.